# Prediction Models of Shielding Effectiveness of Carbon Fibre Reinforced Cement-Based Composites against Electromagnetic Interference

**DOI:** 10.3390/s23042084

**Published:** 2023-02-13

**Authors:** Shilpa Narayanan, Yifan Zhang, Farhad Aslani

**Affiliations:** 1Materials and Structures Innovation Group, School of Engineering, The University of Western Australia, Crawley, WA 6009, Australia; 2School of Engineering, Edith Cowan University, Joondalup, WA 6027, Australia

**Keywords:** shielding effectiveness, cementitious composites, carbon fibre, regression, BPNN

## Abstract

With the rapid development of communication technology as well as a rapid rise in the usage of electronic devices, a growth of concerns over unintentional electromagnetic interference emitted by these devices has been witnessed. Pioneer researchers have deeply studied the relationship between the shielding effectiveness and a few mixed design parameters for cementitious composites incoporating carbon fibres by conducting physical experiments. This paper, therefore, aims to develop and propose a series of prediction models for the shielding effectiveness of cementitious composites involving carbon fibres using frequency and mixed design parameters, such as the water-to-cement ratio, fibre content, sand-to-cement ratio and aspect ratio of the fibres. A multi-variable non-linear regression model and a backpropagation neural network (BPNN) model were developed to meet the different accuracy requirements as well as the complexity requirements. The results showed that the regression model reached an R^2^ of 0.88 with a root mean squared error (RMSE) of 2.3 dB for the testing set while the BPNN model had an R^2^ of 0.96 with an RMSE of 2.64 dB. Both models exhibited a sufficient prediction accuracy, and the results also supported that both the regression and the BPNN model are reasonable for such estimation.

## 1. Introduction

Electromagnetic waves (EMWs) are radiation that are found in the atmosphere in various forms, such as natural light, ultraviolet radiation, radio waves, etc., in the electromagnetic spectrum [1]. They can be generated in the atmosphere by natural occurrences such as lightning or by manmade instruments such as electronics [2]. While most of the waves are created for communication purposes, such as radio waves, some are generated as by-products from these electronic devices and other common devices, such as mobile phones and microwave ovens, etc. This has led to electromagnetic pollution because these waves can ionise the air and induce eddy currents which can cause interference with the functionality of other devices [2]. In the modern world where most operations are dependent on computers and logic systems, an interference of a few volts can disrupt the operation or even cause permanent damage [3] to devices or systems. Previous research by the military has shown that EMI can be caused intentionally in warfare to hack information or even damage critical devices [4]. Apart from these, studies have shown that, at a higher intensity level, EMWs do have negative impacts on pregnant women, foetuses and infants, etc., and prolonged exposure can also cause cancer and other heart-related issues [5].

Most of the early research into EMI and the required amount of the shielding effectiveness (SE) of buildings has been done by the military since they were the most vulnerable to intentional EMI. Apart from military use, since mobile phones and other electronics have gained popularity and become ubiquitous, civil buildings are now also required to provide necessary shielding against EMI [2] to minimise or avoid these negative effects. Traditionally, to provide adequate SE, metals, including copper, steel and aluminium, have been used in the past in hospitals against magnetic resonance imaging (MRI) radiation. However, it also should be noted that these materials are bulky, prone to corrosion and require constant maintenance, which can incur extra costs [1,2]. This, therefore, has led to an increased amount of research into developing innovative construction materials that can provide adequate SE for buildings.

As aforementioned, metals have been commonly used to shield against EMI, but their drawbacks are also obvious, such as being vulnerable to corrosion and having high specific weights, etc. [2]. Moreover, with the increasing demand of high-rise buildings and the modernisation of electronics, buildings require more lightweight, flexible and corrosion-resistant structures. This, therefore, reflects the increasing demand for advanced construction materials with adequate shielding properties. Publications over the past few decades have proposed materials such as polymers, wood-based, modified metallic materials, or cementitious composites to serve the need of the SE since theoretical analysis into the material’s shielding properties has proven that the electrical conductivity and the permeability of the material are critical for shielding effects [6]. Cementitious materials naturally exhibit a slight conductivity due to the ion transfer within the free water in the mixture. Accordingly, studies into developing innovative cement-based construction materials incorporating conductive fillers, which can increase the electrical conductivity and provide sufficient shielding properties, has become burgeoning.

Carbon fibres (CFs) are one of the representatives of such fillers that can be used to enhance the shielding properties of cementitious composites. Its effects and the overall performance of the cementitious composites incorporating CFs have been broadly investigated by pioneer researchers. Some studies have also identified that one of the main advantages of CFs over other fillers, such as carbon nanotubes, steel fibres, etc., is its cost-effectiveness [2]. Apart from cement-based composites, a group of researchers have also attempted including CFs with different sizes into geopolymer composites and studied the overall shielding properties of such CF-reinforced geopolymers [7]. The study tested the shielding properties of the composites incorporating CFs with lengths of 3, 6 and 12 mm at weight percentages of 0.1%, 0.3%, 0.5% and 0.7% from 30 MHz to 1.5 GHz and reported that increasing either the length or the content of the fibres contributed to the improvement of the overall SE of the composites. However, the growth of the improvement showed a descending trend, meaning there would be a saturation limit of increasing the SE. However, some researchers are also concerned that the sole conductive additive may not provide sufficient improvement to the SE to meet the industrial standard requirements, and have introduced a series of second additives that have been reported to have potential to further increase the SE. Wanasinghe et al. studied the overall shielding properties of the composites incorporating different types and contents of carbon nanofibres, with varied fractions of zinc oxide and activated carbon powder combined with 12 mm unsized carbon fibres following the standard of ASTM D4935-18 [8]. Their results revealed that the inclusion of the activated carbon powder showed the most significant improvement in electrical conductivity as well as the overall SE among all the types of additives. However, the authors also reported an issue of increasing ACP content up to 2%, in which the mixes became crumbled and unworkable [8].

## 2. Research Significance

With the growing awareness of the importance of material shielding properties, plenty of researchers have been exploring the behaviour and mechanisms of the EMI shielding properties of the composites subjected to external electromagnetic waves through either conducting physical experiments or performing numerical modelling. To estimate the SE of cementitious composites or concretes, one of the commonly used models is the Debye model, which was first introduced by Peter Deby in 1929 [9] and is used to describe the frequency behaviour of the permittivity of the materials. Its frequency-dependent complex relative permittivity obeys Equation (1) [10]:(1)$$\hat{\varepsilon_{r}}\left(\omega \right)=\varepsilon_{r}^{\prime }\left(\omega \right)-j\varepsilon_{r}^{\prime \prime }\left(\omega \right)\;\;\;\;\;\;\;\;=\varepsilon_{\infty }+\frac{∆\varepsilon }{1+j\omega \tau }\;\;\;\;\;\;\;\;\;\;\;\;\;\;\;\;\;\;\;\;\;\;\;=\varepsilon_{\infty }+\frac{∆\varepsilon }{1+\omega^{2}\tau^{2}}-j\frac{\omega \tau ∆\varepsilon }{1+\omega^{2}\tau^{2}}$$
where ω is the frequency ranging from zero to infinity; ∆ε = $\varepsilon_{static}-\varepsilon_{\infty }$ is the difference between the values of the real part of the complex relative permittivity at a low and high frequency, respectively, and τ is the relaxation time. Researchers then extended and developed plenty of frequency-dependent models such as the Cole–Cole equation, the Jonscher model, etc. [10,11,12]. However, it should be noted that, apart from the frequency, both of these models require relaxation time, which is required to be measured directly or determined indirectly. Based on this, this study aimed to study the relationship between the SE of concrete and cementitious composites by direct measurement and mixed design parameters affecting the SE of composites, then develop and propose a series of mathematical models that can estimate the SE by using the mixed design parameters without involving the other indirect variables. Moreover, from the consideration of the diversity of the available materials, the varied impact of the different materials on the SE, as well as the scope and limit of a single research paper, the study was then narrowed to the composites incorporating carbon fibres. It should also be noticed that the proposed model should be compared with the existing models from the previous literature to demonstrate its superiority and progression. However, since the available models from the previous literature are mainly focused on the extension or progression of relaxation-time-based models, which do not involve mixed design parameters, it should be noted that it is unfair to compare these models with the models to be proposed due to the different input parameters as well as the different prediction methods.

It should be noticed that, although the results from the previous literature have proven that the presence of conductive fibres significantly improves the composites’ overall SE, most of the high-conductive fibres, such as CFs, are costly and incur higher capital costs for the overall project [2]. Moreover, conducting such experiments for the shielding properties is also expensive and time consuming compared to numerical modelling. The previous literature has shown that the interaction between the materials’ properties and machine learning algorithms has grown rapidly, and the good performance of machine learning algorithms in estimating or predicting these properties using mixed design parameters has been witnessed by a wide range of research articles [13,14,15,16,17]. Machine learning models have a series of advantages, such as time saving and cost saving, over conducting physical experiments. With these, this study aims to develop and propose a series of prediction models that can be used to estimate the EMI SE of such cementitious composites containing CFs at a given frequency by using mixed design parameters. For this purpose, a multi-variable non-linear regression model and a backpropagation neural network model (BPNN) were then developed to meet the different prediction accuracy requirements and complexity. To meet such aims, a database containing 346 pairs of data from 10 different studies was complied.

## 3. Background of the Shielding Effectiveness of Cement Composites

### 3.1. Shielding Effectiveness

EMWs are represented as a sinusoidal pair of electric and magnetic fields vibrating perpendicular to each other. These waves are characterised by their wavelength, which is the distance between the two consecutives peaks or, accordingly, their frequency (*f*). The equation relating the energy (*E*) within a wave and its frequency is shown in Equation (2):(2)E=h×f 
where *h* is the Planck constant.

Hence, these EMWs have different energies based on their frequencies. Classically, the EMWs are divided into different spectrums based on their energies. The waves at the lower end of the spectrum have lower frequencies, i.e., lower energies and so on [2]. When a EMW falls on a material, there can be three types of effects: absorption, reflection or transmission (through the material) as shown in Figure 1. When the incident wave intensity is reduced upon transmission, it is called shielding [2]. Therefore, the total SE is represented as Equation (3), where SE_A_ is due to the absorption, SE_R_ is due to the reflection and *SE_M_* is due to multiple reflections:(3)$$SE=SE_{A}+SE_{R}+SE_{M}$$

The reflection occurs due to the impedance mismatch between the incident waves and the surface of the material [18], which can be expressed as Equation (4) where *f* is the frequency, *ϵ* is the electrical permittivity, *μ* is the magnetic permittivity and *σ_T_* is the total electrical conductivity:(4)$$SE_{R}=-10\log\left(\frac{\sigma_{T}}{16f\epsilon \mu_{r}}\right)\;$$

Hence, the electrical conductivity plays a major part in the shielding. Materials with higher conductivity, such as metals, are good reflectors [2]. Materials with multiple fillers can undergo multiple internal reflections due to the different dielectric properties of each filler. The SE due to multiple reflections is shown in Equation (5), where *δ* is the skin depth of the material [1]:(5)$$SE_{M}=20\log\left(1-e^{\frac{-2z}{\delta }}\right)$$

EMWs can undergo absorption, which is dependent on the dielectric properties of the material. The SE due to absorption can be represented as Equation (6), where *α* is the attenuation and *z* is the thickness of the material [1]:(6)$$SE_{A}=20\log\left(\frac{1}{e^{-\alpha z}}\right)$$

The SE can also be defined as the ratio of the received power of the beam with the material present ($P_{1}$) to the received power of the beam without the material present ($P_{2}$) [1], as shown by Equation (7):(7)$$SE=10\log\left(\frac{P_{1}}{P_{2}}\right)\;$$

### 3.2. Mix Design Parameters Affecting Shielding Effectiveness

Cementitious composites are one of the most commonly used materials for conductivity and have been the subject of many studies which point out that the conductivity and SE of the composites depend on many factors, such as the type of additives, water-to-cement ratio, porosity and fillers, etc. [19].

To investigate the effect of the water-to-cement ratio on the shielding properties of the cementitious material, Wanasinghe et al., therefore, conducted a series of experiments to observe the changes in the SE of mortar within a frequency band of 30 MHz to 1.5 GHz [6]. Their results show that increasing the water-to-cement ratio led to a decrease in the SE of the mortar within the given band, while increasing the water-to-cement ratio made the specimen more porous and decreased the specimen density, hence adversely affecting the ability of the EMI absorption of the specimen. It also should be noted that a more porous structure would also have higher multiple reflections. However, this enhancement was not adequate to compensate for the loss of the overall EMI SE by absorption and reflection. The study also claimed that the optimum water-to-cement ratio that yielded the best EMI SE was 0.3 [6].

It was established that, with the increase in the fibre content in the cement composite, the SE was shown to increase [20], with the maximum transmission SE produced by the largest amount of CFs [1]. It was also shown that, with a higher CF content, the electrical conductivity of the composite was seen to increase due to the conductive network created [1]. Their results also revealed that the specimens containing unsized CFs generally showed an improvement in the SE of the composites compared to the sized fibres. They also claimed that the mix design which showed the relatively best SE of 40–60 dB from 300 MHz to 1.5 GHz was the one incorporating the 12 mm unsized CFs at a fibre fraction of 0.7 wt%. Similarly, Zhang et al. conducted a series of experiments using steel fibre, carbon fibre and polyvinyl alcohol fibre (PVA) at varied volume contents from 0.5% up to 3% [21] with a frequency band of 80 MHz to 10 GHz. Their results showed that, for both fibres, increasing the fibre content increased the overall SE.

Past studies have shown that there is an effect of the sand ratio on the EMI SE. The addition of sand has been shown to decrease the resistivity of the material [22]. However, studies into the effect of the sand content on the shielding properties of composites or concrete is still limited and, thus, shows a potential for further investigation. Moreover, research shows that an increase in the thickness of the material increases the interaction between the material and the EMW, causing a higher absorption of the EMW [6,23] since the thickness also impacts the skin depth while the frequency plays a major part in deciding the effect of the thickness on the attenuation of the waves. The aspect ratio is the ratio of the length to diameter of the fibre. In addition, the previous literature has observed that the aspect ratio of the fibre also plays a role in affecting the SE of the composites considerably [1,2,20]. An experiment by Wanasinghe et al. (2020) showed that the transmission of the SE produced by the 3 mm fibres and 6 mm fibre mixed specimens generated significantly higher SE in comparison to the fibres with the same diameter [1]. The addition of fibres with longer lengths led to a larger interconnected conduction network, thereby increasing the overall SE [1]. Wang et al. [24] studied the effects of CFs on the shielding properties of composites subjected to environmental conditions. This study explained the behaviours of the SE of the composites containing CFs during freezing and thawing cycles. Their results showed that within a band of 2–18 GHz, freezing and thawing did not show obvious impacts on the SE for the composites not containing CFs. After freezing and thawing, the composites containing CFs revealed an increase in the EMW reflection while the absorption decreased at a high frequency.

## 4. Experimental Database

Based on the results and conclusions by the previous literature, the parameters, including the fibre content, water-to-cement ratio (by mass), sand-to-cement ratio (by mass), aspect ratio of fibres, specimen thickness and frequency, were reported to have significant effects on the shielding effectiveness and were selected for further analysis. A database containing 346 pairs of data from ten different literature sources was then complied and the variables collected included the fibre content, water-cement ratio (w/c), sand-cement ratio (s/c), aspect ratio of fibre, specimen thickness, frequency and SE. Table 1 summaries the key statistical results of the database and detailed database can be viewed in Appendix A. It should be noted that all the data were not normalised for the model development.

### 4.1. Fibre Content

Figure 2 presents the scatter plots of the SE against the CF content in volume percentage with colour mapping in the frequency of the fibre content from (a) 0–4%, (b) 0–1% and (c) 2–4%, respectively. The plot reveals that the majority of the data fell within a range of 0–1%, as only five pairs of data were beyond 1%, indicating that the database needs to be enriched in the future with more research outcomes into the shielding properties of carbon-fibre-reinforced composites at a high fibre content, especially at a high frequency. From the plot in Figure 2c, for the points containing CFs beyond 1 vol%, it can be seen that, with an increase in the fibre fraction, the overall SE shows a general growing trend up to a percentage of 4 vol%. This observation is consistent with the findings reported by Wanasinghe et al. [1] and Zhang et al. [21] that increasing the fibre content leads to an increase in the overall SE. However, it should be emphasised that Figure 2b does not indicate a significant trend that increasing the fibre content leads to an increase in the overall SE. It might be since the EMI shielding property is governed by three different mechanisms as described previously. The interference of the different mechanisms might lead to undesirable behaviours or changes [1]. Additionally, factors such as the changing porosity and subsequent permeability, impedance mismatch and surface conditions of the cement matrices could also affect the overall SE significantly [29]. Moreover, the tendency that Figure 2b remains unclear might also be attributed to other governing factors, such as the water-to-cement ratio, fibre type, specimen thickness, etc., and that the single plot could not eliminate the effects of the other parameters unless it performed the control variates method in future study.

### 4.2. Water-to-Cement Ratio

Research has shown that the water-to-cement ratio has a considerable effect on the EMI SE of the cement composites [6]. Increasing the water content increases the porosity, which increases the multiple reflection of the EMW within the material. The presence of fillers or fibres aids this phenomenon. It is interesting to note that, with a higher porosity, the density of the specimen decreases and, thus leads, to a lower SE. Figure 3 reveals that increasing the water-to-cement ratio will obviously worsen the overall SE of the composites, and it is consistent with the conclusions from the previous literature [6]. The figure also illustrates that the SE increases with an increase in the frequency.

### 4.3. Sand-to-Cement Ratio

Past studies have shown that there is an effect of the sand ratio on the EMI SE, as the addition of sand has been shown to decrease the resistivity of the material [22]. From Figure 4, it can be witnessed that, with an increase in the sand-to-cement ratio, the plot generally exhibits a slight decrease when the ratio reaches approximately 1 followed by a significant increase. It should also be noticed that due to the limit of lacking a diversity of data in the sand-to-cement ratio, this trend might not be adequate to propose a general trend of changing behaviours. As discussed in Section 3.2, the study on the effects of the sand content on the shielding effectiveness is still limited, and the mechanism behind this trend might be unclear and complicated. Moreover, this also reflects the potential for studying the influence of the changing sand content on the overall shielding effectiveness for pioneer researchers.

### 4.4. Thickness of Specimen

According to Figure 5, it has been witnessed that the data are mainly located at the thickness of approximately 10 mm and 100 mm and showed a big blank in between. It is mainly attributed to the differences between the testing standards and most of the data gathered used a similar specimen geometry. It should be noted that it is brutal to conclude that increasing the specimen thickness had a higher overall SE due to a lack in the variation in the data for the specimen thickness. However, the thickness of the material also plays a significant role in generating a high EMI shielding effectiveness in a material [23]. This therefore reflects and indicates the possibility of further research into considering size effect on the SE.

### 4.5. Aspect Ratio

From Figure 6, it can be seen that with a higher frequency, the SE is higher. Within a frequency band of 0–6.225 GHz, the increase in the aspect ratio tends to decrease the SE and then slightly increase.

## 5. Prediction Models

In this study, two types of prediction models for estimating the EMI SE of the composites incorporating carbon fibres were developed, namely a multi-variable non-linear regression model and a backpropagation neural network model (BPNN) based on the Levenberg–Marquardt algorithm. The multi-variable non-linear regression model developed was based on the ordinary least squares method, and it aimed to be able to estimate the SE of a composite at a given frequency using mixed design parameters with a certain accuracy and simpler mathematical expression. The BPNN model was designed to be applicable for more accurate predictions compared to the least square regression model, but with a more complex expression in the matrix format. To develop such models, the database established was split into training, validation and test sets by random selection from each literature at a ratio of 70–15–15% as a common ratio. However, it should be noted that, from the consideration of the limited pairs of data gathered, the leave-one-out cross-validation method was then chosen to validate the BPNN model to avoid the validation set. The database was therefore split into training and test sets with a ratio of 70–30%. Both the regression model and the BPNN model were then developed based on the training set and their performances were evaluated by the test set. All the models were developed using the MATLAB platform.

### 5.1. Regression Model

As discussed in Section 4, six parameters were considered as the inputs for the regression model, namely the fibre content in the volume percentage, water-to-cement ratio by mass, sand-to-cement ratio by mass, thickness of the specimen, aspect ratio of the fibre and frequency of the target output for the EMI SE in dB. By tracing the raw data, it was observed that all the parameters were linearly proportional to the EMI SE, either positivity or negatively, although there was a wide scatter due to the interdependency of the individual parameters. Therefore, the following regression model was proposed as
(8)$$SE\prime =b_{1}+b_{2}*v+b_{3}*w+b_{4}*s+b_{5}*z+b_{6}*a+b_{7}*f$$
where $SE^{\prime }$ is the predicted SE, *v* is the fibre volume percentage, *w* is the water-to-binder ratio, *s* is the sand-to-cement ratio, *z* is the thickness of the specimen, *a* is aspect ratio, *f* is the frequency of the EMW and $b_{1}$ to $b_{7}$ are the regression coefficients. To obtain the regression coefficients, the ordinary least square (OLS) regression method or simply least squares method was used. The regression coefficients obtained are shown in the following equation and the proposed model had an RMSE of 2.3 dB with an R^2^ of 0.88. These indicated that the model was a good attempt for estimating the SE at a given frequency with mixed design parameters and has the ability to explain the data.
(9)$$SE^{\prime }=-730.81+6.47v+691.49w+380.42s+1.078z-0.0071a+2.87f$$

### 5.2. BPNN Model

A backpropagation neuron network model (BPNN) is one type of an artificial neuron networks (ANN), which assesses the differences between the predictions and labels, then propagates the errors to each neuron to adjust the weightings and biases to achieve the expected training goal. BPNN models or neural network-based models have been deeply studied and widely applied in predicting or estimating fresh or hardened properties of concrete or cementitious composites [30,31,32,33]. A simple BPNN model consists of three layers: an input layer, numerous hidden layers and an output layer, as shown in Figure 7. The connections between the neurons in the different layers can be expressed as
(10)$$y_{j}=f\left(\overset{n}{\underset{j=1}{\sum }}\left(w_{ji}x_{i}+b_{ji}\right)\right)$$
where *w_ji_* is the weighting between neuron *i* in the upper layer and neuron *j* in lower layer, *x_i_* is the output from neuron *i*, *b_ji_* is the bias between neurons *i* and *j*, *f* is the activation function (transfer function) that is used to map the inputs from the previous layer to a given range and *y_j_* is the output from neuron *j* [14]. In this paper, the activation function selected for the hidden layer is a tangent sigmoid function as
(11)$$f\left(x\right)=\frac{2}{1+e^{-2x}}-1$$

In the training process, the iteration loop was broken when the mean squared error (MSE) was less than the pre-defined training goal and the training stopped.
(12)MSE=1n∑i=1n(yi−yi^)2
where *y_i_* and ${\hat{y}}_{i}$ are the predictions from the model and labels, respectively [14].

In this study, with the consideration that the size of the database complied was limited, the number of input and output variables were simple. Since the idea was to keep the model as simple as possible for user friendliness, the BPNN model was designed to have a simple architecture. As discussed earlier, the dataset was divided into training and testing sets with the ratio of 70–30%. The parameters to be pre-configured were the basic training parameters and the hyperparameters. The training parameters, such as a training function, epochs, learning rate and learning goal, etc., decided the training process, and the hyperparameters defined the structure of the network. The training function was selected to be the default learning function based on the Levenberg–Marquardt algorithm [14]. Using this training function, the validation process was performed using the data from the training set to tune the other hyperparameters. Due to this, the leave-one-out cross-validation method was considered to achieve the hyperparameter tuning, following which the validation set was combined with the training set. In this study, a single hidden layer model was selected due to the complexity of the problem and the hyperparameters to be tuned was a number of hidden neurons in this layer. In this case, the hidden neurons ranging from one to 10 were validated and each model was repeated 10 times to take average root mean squared error (RMSE). The results are summarised in Table 2. The result show that the model with two neurons had the minimum average RMSE and, hence, this was selected.

In this study, the model’s performance was assessed based on two parameters, namely the root mean square error (RMSE) and the coefficient correlation (R), which describe the degree of the correlation of the input parameters to the SE. The proposed BPNN can be mathematically described as Equation (7). Due to the heavy computation, we attempted to minimise the computation required by mapping the inputs from 0 to 1 according to the following equations. It should also be noted that the output of the model was mapped back for a result discussion, as shown in Equations (8)–(10). The developed BPNN model offered an R and RMSES of 0.962 and 0.31 dB for the training set. For the test set, the model reached an R and RMSE of 0.983 and 2.64 dB, respectively, as shown in Figure 8. These indicate that the proposed BPNN model has the ability to estimate the SE at a given frequency using mixed design parameters at a relatively good level.
(13)$$\begin{array}{l} SE=tansig\left(⎣\begin{array}{ccc} 1.443 & -2.049 & 0.274\\ 0.461 & -0.769 & -3.116 \end{array}\;\begin{array}{ccc} 3.154 & -2.578 & 6.874\\ -0.192 & -0.698 & -0.549 \end{array}⎦*\left|\begin{array}{c} \begin{array}{c} \%vol\\ w/c\\ s/c \end{array}\\ thickness\\ aspRatio\\ freq \end{array}\right|+\left|\begin{array}{c} -2.782\\ 3.593 \end{array}\right|\right)\\ \;\;\;\;\;\;\;\;*\left(\left|\begin{array}{cc} \;6.062 & -0.4057 \end{array}\right|\right)-0.516 \end{array}$$
(14)$$Input\;mapping\;:\;\left[\begin{array}{cc} Min & 0.5\;\;\\ Max & 4 \end{array}\begin{array}{c} \;\;0.3\\ \;0.5 \end{array}\begin{array}{cc} \;\;\;1 & 3.9\\ \;\;\;1.2 & 100 \end{array}\begin{array}{cc} \;\;\;100 & 0.7\\ \;\;\;\;625 & 12.44 \end{array}\right]$$
(15)$$Output\;mapping\;:\;\left[\begin{array}{cc} Min\; & 15.5\\ Max & 68.13 \end{array}\right]$$
(16)$$\mathrm{The}\;\mathrm{equation}\;\mathrm{for}\;\mathrm{mapping}:\;y=\left(y_{max}-y_{\min}\right)*\frac{x-x_{\min}}{x_{max}-x_{\min}}+y_{\min}$$

## 6. Conclusions

This study proposed two prediction models, a multi-variable non-linear regression model and a BPNN model, to meet the different accuracy requirements with various complexity for estimating the shielding effectiveness of cementitious composites containing carbon fibres at a given frequency using mixed design parameters, including the fibre fraction, water-to-binder ratio, sand-to-cement ratio, thickness of specimens and fibre aspect ratio. The conclusions of this study can be summarised as follows.

The data aggregation revealed that the fibre fraction of 0–1 vol% was commonly used since the majority of the data fell within this range. At each fibre content, it was clear to see the increase in the SE with a higher frequency as expected. With an increase in the fibre content, the SE did not exhibit a significant increasing trenddue to the interactions of different mechanisms. This behaviour is opposite to the finds being reported in previous literature.The scatter plot for the SE against the water-to-cement ratio illustrated that a higher w/c ratio led to a poorer SE, since a higher water content increased the porosity of the material and, hence, increasing the multiple reflection of the EMW and the inclusion of the fibres aids this deterioration. This observation is consistent with the conclusions drawn in the previous literature.Although the data aggregation gave an indication of the increasing sand-to-cement content, the SE experienced a reduction and then started to increase. It should be noted that there was a lack of data in the sand-to-cement ratio in this database and it showed a potential for pioneer researchers to comprehensively investigate the behaviour of the SE with varied sand content.Conclusions from the previous literature emphasised the non-negligible effect of the specimen thickness on the shielding effectiveness. However, the data aggregation did not indicate sufficient evidence that proved the correlation between the specimen thickness and the material’s SE due to the limited data in the thickness. Further research is required to comprehensively investigate the effect of the specimen thickness and corresponding size effect on the shielding effectiveness.The water-to-binder ratio was negatively related to the SE unless the conductivity was large enough to cause an effect to reverse. This effect may have been noticed since the relation was positive between the SE and the water-to-binder ratio.The multi-variable non-linear regression model was deemed reasonable to estimate the EMI SE. It was confirmed by the high R^2^ value of 0.88 and low RMSE of 2.3 dB. The BPNN model also offered a good prediction accuracy since its best R^2^ and RMSE reached 0.96 and 2.64 dB.

Although the parameters, such as the sand-to-cement ratio, specimen thickness and aspect ratio of the fibres, were reported to have a non-negligible effect on the overall shielding effectiveness of the concrete or cementitious composites, the results from the data aggregation did not exhibit the expected behaviour of the SE and the mechanisms or correlations behind the effects of these factors on the SE still remained unclear due to the built-in disadvantage of being unable to explain the physical theory behind the predictions for the regression or machine learning. These, therefore, reflect the possibility for pioneer researchers to further investigate the behaviour of the SE with these parameters by conducting physical experiments. Moreover, since the BPNN model as well as the regression model exhibited a reasonable performance, it is possible for pioneer researchers to include the data of the other additives or fillers instead of solely carbon fibres in the database and develop prediction models with a higher ability of generalisation. Apart from the BPNN model, there are plenty of models available from machine learning, and they also have the potential for combining different classifiers to form a hybrid system or boost the system to further overcome the disadvantages of the BPNN model, thus improving the general performance of the model.

## Figures and Tables

**Figure 1 sensors-23-02084-f001:**
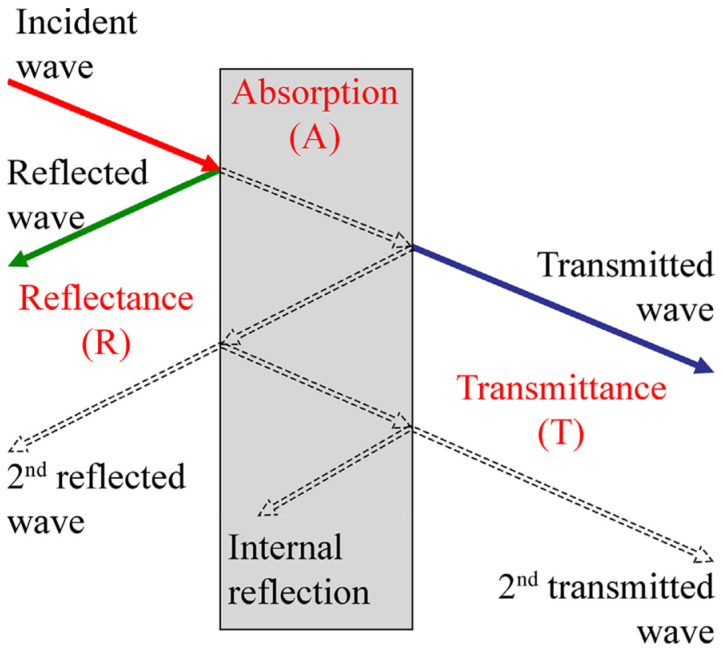
Possible interaction of an EMW with a material [2].

**Figure 2 sensors-23-02084-f002:**
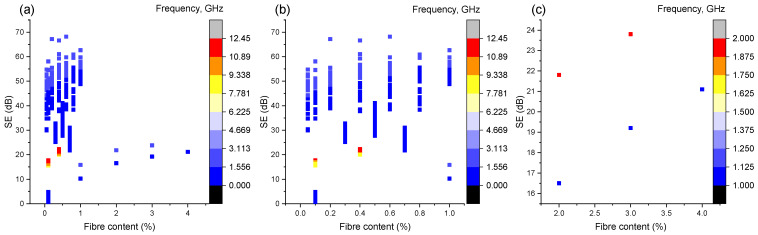
Scatter plot of the SE vs. the fibre content at (**a**) from 0–4%; (**b**) 0–1%; (**c**) 2–4%.

**Figure 3 sensors-23-02084-f003:**
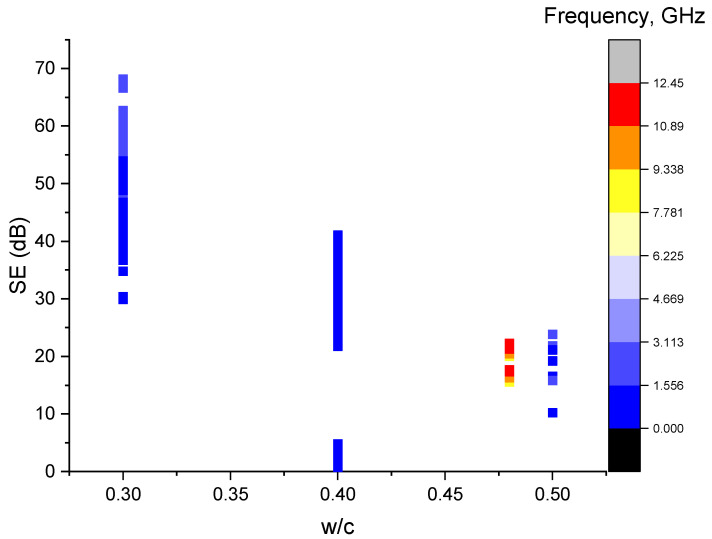
Scatter plot of the SE vs. the water-to-cement ratio.

**Figure 4 sensors-23-02084-f004:**
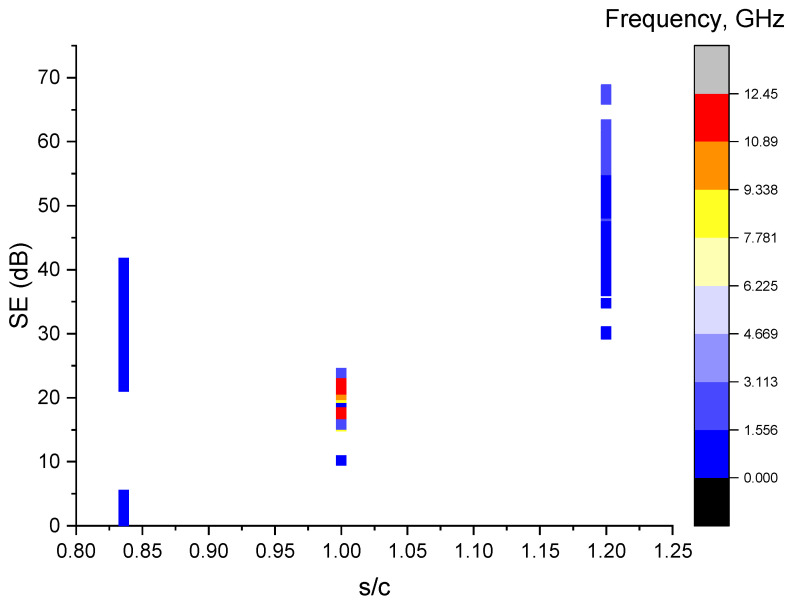
Scatter plot of the SE vs. the sand-to-cement ratio.

**Figure 5 sensors-23-02084-f005:**
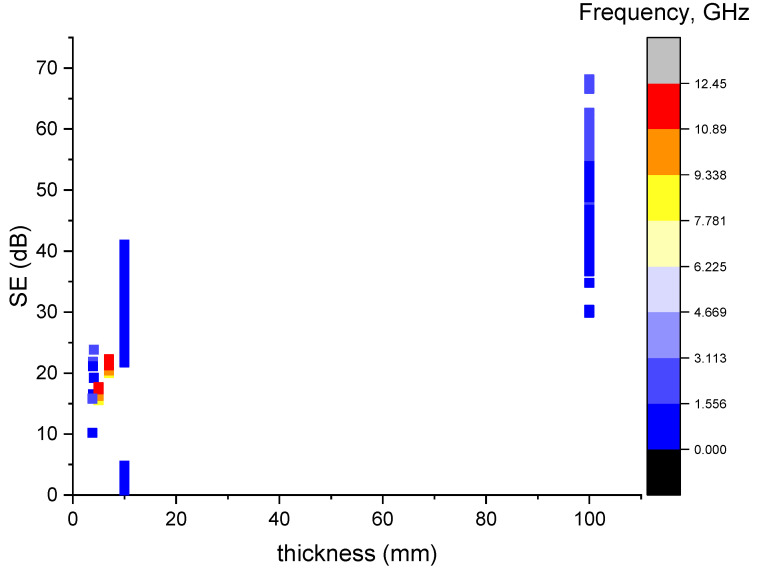
Scatter plot of the SE vs. the specimen thickness.

**Figure 6 sensors-23-02084-f006:**
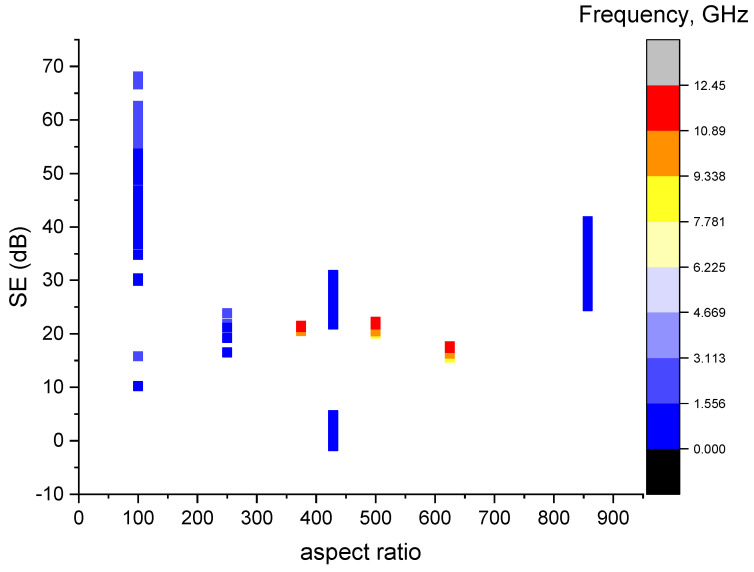
Scatter plot of the specimen aspect ratio vs. the SE.

**Figure 7 sensors-23-02084-f007:**
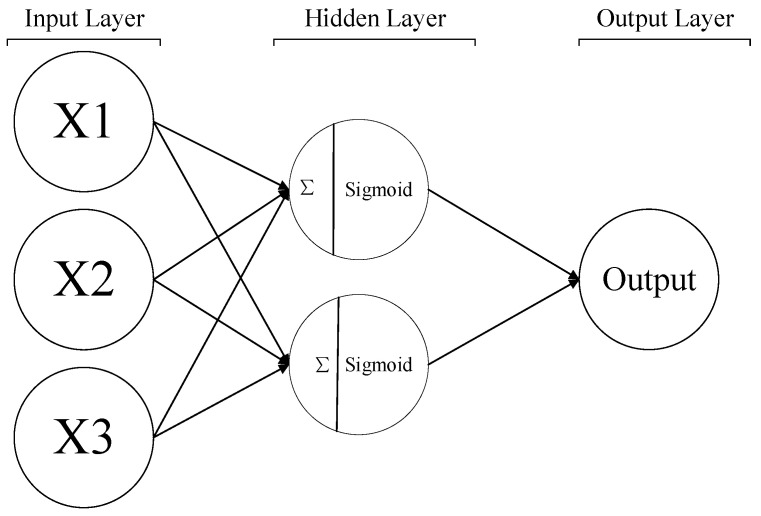
Structure of a simple BPNN [14].

**Figure 8 sensors-23-02084-f008:**
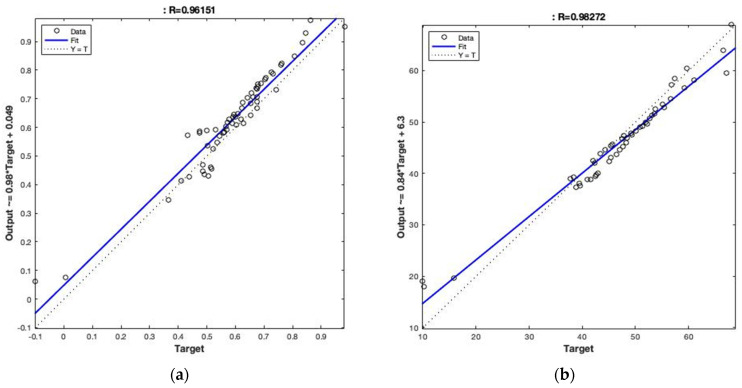
Training and test results of the BPNN model (**a**) training; (**b**) test.

**Table 1 sensors-23-02084-t001:** Statistics of the database [1,2,6,20,21,25,26,27,28].

Variable		Unit	Min.	Max.	Mean
Fibre, %	Input		0.05	4.00	0.27
w/c	Input		0.20	0.50	0.38
s/c	Input		0.80	1.20	1.05
Aspect ratio	Input		250	857	322.88
Specimen thickness	Input	mm	3.80	100	52.16
Frequency	Input	GHz	0.00090	12.44	3.37
SE	Output	dB	10.20	68.13	37.04

**Table 2 sensors-23-02084-t002:** Cross-validation results.

Neurons	1	2	3	4	5	6	7	8	9	10
Avg RMSE	0.444	0.391	0.688	0.593	0.519	0.612	0.591	9.531	0.527	3.765

## Data Availability

Not applicable.

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
