# Peer review of "Prediction Models of Shielding Effectiveness of Carbon Fibre Reinforced Cement-Based Composites against Electromagnetic Interference"

_sensors, 2023, doi:10.3390/s23042084_

Round 1
Reviewer 1 Report
1- Abstract
To increase the quality of the article, the abstract must undergo extensive rewriting.
2- Introduction
In general, the literature review section of the introduction is poorly arranged. The introduction requires revision.
3- The significance of the study must be described in the separated section after the introduction.
4- The input parameters listed in Table 1 were selected based on what factors?
5- How are the available data sets used to create the training and testing data sets? Certain previous studies suggest a methodology for dividing the entire database into training, validation, and testing subsets, which should be highlighted in the study.
6- The results of previous studies have shown that as the number of layers increases, the system's complexity increases. What is the reason for choosing one hidden layer?
7- What are the criteria of accepting the idealized network?
8- This study does not compare existing relationships. As a result, readers are unable to determine how the presented model compares to existing design methods. The reviewer suggests including the comparison.
9- The authors need to support the following sentences with appropriate references.
9-1- Waves at the lower end of the spectrum have lower frequencies, i.e., lower energies and so on.
9-2- Materials with higher conductivity like metals are good reflectors.
9-3- Backpropagation neuron network model (BPNN) is one of the types of artificial neuron network (ANN), which assesses the differences between predictions and labels, then propagates the errors to each neuron to adjust weightings and biases to achieve the expected training goal.
the following references are related to your work:
https://doi.org/10.1016/j.matpr.2021.03.263
10- The paper presents a large number of results but without a theoretical and practical approach. Also, the reviewer believes that the results part reports only the differences in the observation without detailed descriptions of reasons and stays quite superficial.
The results section should be completely rewritten.
It is possible to compare results in an efficient manner.
11- Conclusions
The conclusion section needs to be re-written by incorporating general conclusions from the findings of this research.
12- The descriptions of the axis and legend in different figures are not of the same size fonts. Furthermore, the authors can enhance the quality of the figures.
Author Response
- To increase the quality of the article, the abstract must undergo extensive rewriting.
The authors thank for this comment. The abstract has been revised based on all the comments received from both reviewers. Please see line 9 – 22.
- In general, the literature review section of the introduction is poorly arranged. The introduction requires revision.
The authors thank for this comment. The introduction has been re-organised and revised based on all the comments received from both reviewers regarding this section. A few more references have also been added to this section. Please see line 24 – 86.
- The significance of the study must be described in the separated section after the introduction.
The authors thank for this comment. The research significance of this paper has been added and emphasised in section 2. Research significance. Please see line 76 – 126.
- The input parameters listed in Table 1 were selected based on what factors?
The authors thank for this comment. The parameters selected are reported to have non-negligible impact on shielding effectiveness of cementitious composites. Please see line 155 – 199.
- How are the available data sets used to create the training and testing data sets? Certain previous studies suggest a methodology for dividing the entire database into training, validation, and testing subsets, which should be highlighted in the study.
The authors thank for this comment. In terms of machine learning models, it is fairly normal to divide the entire database into training, validation, and test sets at a common ratio of 0.7-0.15-0.15. However, as what has been mentioned in either this paper or many other papers which are associated with machine learning algorithms, this ratio might be adjusted based on the size of the database. With regard to this paper, the database established is not a big database compared to those big data models. Considering this, to boost the performance of the model and minimise subsequent errors, the authors decided to merge training set and validation set at a overall ratio of 0.7 and use Leave-one-out method to validate the models using such merged set.
- The results of previous studies have shown that as the number of layers increases, the system's complexity increases. What is the reason for choosing one hidden layer?
The authors thank for this comment. The authors agree with the statement of system’s complexity was increased by increasing the number of either overall layers or hidden layers. However, the authors believe that more layers bring not only much more complexity of a system as well as subsequent much heavier computation work, but also less applicability for peoples who are not familiar with machine learning algorithms. Moreover, a model or system with more layers will suffer from the problems of overfitting. With these considerations, the authors developed two types of models, regression and BPNN model, aiming to address different accuracy requirements by different users with different complexity and applicability. The reason of using single hidden layer for this model is firstly, from the same idea of keeping model as simple as possible; secondly, the number of input parameters involved in this paper is not complicated and more layers might lead to the problem of overfitting.
- What are the criteria of accepting the idealized network?
The authors thank for this comment. In this study, the network was evaluated and selected based upon R2 value as well as root-mean-square error (RMSE). As the authors mentioned in the context, due to the limit of the size of the database, the model was trained and validated using leave-one-out cross-validation method at a training set ratio of 0.7 (or 70%). However, with the consideration of problem of randomly initialising weightings and biases from MATLAB before training as well as the problem of global minimum and regional minimum while descending, the authors decided to use average RMSE for 10 repeats as an indication of general performance of networks. After training, the model with minimum averaged RMSE was selected and then proceeded to testing.
- This study does not compare existing relationships. As a result, readers are unable to determine how the presented model compares to existing design methods. The reviewer suggests including the comparison.
The authors thank for this comment. The authors agree with the idea of comparing proposed model with available models or approaches from previous literature to demonstrate the bottleneck in this filed and prove the novelty of this manuscript. However, after literature survey, such existing models that is able to give estimation for SE using mix design parameters were not founded in current database and this might reflect the novelty of this paper of using such regression and BPNN models as well as mix design parameters to predict SE for carbon fibre reinforced composites. Besides, the existing models are mainly associated with relaxation time, which was not intended to be included in this study.
- The authors need to support the following sentences with appropriate references. Waves at the lower end of the spectrum have lower frequencies, i.e., lower energies and so on. Materials with higher conductivity like metals are good reflectors.
The authors thank for this comment. Reference for these two sentences has been added. Please see line 134 -135 and 143 – 144.
- Backpropagation neuron network model (BPNN) is one of the types of artificial neuron network (ANN), which assesses the differences between predictions and labels, then propagates the errors to each neuron to adjust weightings and biases to achieve the expected training goal.the following references are related to your work:https://doi.org/10.1016/j.matpr.2021.03.263
The authors thank for this comment and sharing the article. Considering the growth of interactions of machine learning and studies into materials’ properties, there is a wide range of available research articles on this topic. The authors have added this paper and a few other related papers to this manuscript. Please see line 116 – 119.
- The paper presents a large number of results but without a theoretical and practical approach. Also, the reviewer believes that the results part reports only the differences in the observation without detailed descriptions of reasons and stays quite superficial.
The authors thank for this comment. The results interpretation for data aggregation has been revised and extended in section 3.2. Please see line 155 – 274.
- The results section should be completely rewritten.
The authors thank for this comment. Results section including database presentation as well as model presentation and results have been re-organised, revised and extended. Please see line 155 – 348.
- It is possible to compare results in an efficient manner.
The authors thank for this comment. Figures involved have been re-generated and results discussion has been revised and extended according to all the comments received from all reviewers. Please see line 155 – 358.
- The conclusion section needs to be re-written by incorporating general conclusions from the findings of this research.
The authors thank for this comment. The key findings of this study have been summarised and rewritten according to all comments associated with. Please see line 359 – 408.
- The descriptions of the axis and legend in different figures are not of the same size fonts. Furthermore, the authors can enhance the quality of the figures.
The authors thank for this comment. Some of these unproper figures have been removed all figures used in this manuscript have a minimum resolution of 300 dpi and are in vector form.
Reviewer 2 Report
The authors have established an experimental database by collecting data from previous literature for the shielding effectiveness of carbon fibre-reinforced composites within a given frequency band, and also developed a series of prediction models which is capable to estimate the shielding effectiveness of a mix design at a given frequency using mix design parameters.
The investigated work is quite interesting. Here are some comments to improve the quality of the manuscript.
· ·The start of the abstract is too long. Shorten the abstract (lines 9-18).
· Page 2, line 28: Use the full name form of “RMSE” for the first time.
· Page 2, line 37: It seems that the word “lightening” has a typing error and should be “lightning”.
· Add some references recently published in MDPI journals to the manuscript to enrich the introduction section.
· Add reference for the equations (4, 5, ...) where needed.
· Page 7, line 162: Move the references to the title of table 1.
· Specify the type of fibers in table 1.
· Page 7, line 171: there is a typo. The "fire" must be fiber.
· Add more details to the paper about how you have reached the formula in equation 8.
· Equation 8; modify the sign "±". It must be either "+" or "-"
Author Response
- The start of the abstract is too long. Shorten the abstract (lines 9-18).
The authors thank for this comment. The abstract was revised based on all the comments received from both reviewers. Please see line 9 – 22.
- Page 2, line 28: Use the full name form of “RMSE” for the first time.
The authors thank for this comment. The abbreviation of RMSE has been corrected to root mean squared error and correction has been highlighted in the context. Please see line 19 – 21.
- Page 2, line 37: It seems that the word “lightening” has a typing error and should be “lightning”.
The authors thank for this comment. The typo has been corrected and please see line 27- 28.
- Add some references recently published in MDPI journals to the manuscript to enrich the introduction section.
The authors thank for this comment. The introduction has been re-organised and revised based on all the comments received from both reviewers regarding this section. A few more references have also been added to this section.
- Add reference for the equations (4, 5, ...) where needed.
The authors thank for this comment. The reference for equation 4, 5, and 6 has been added. Please find line 145 – 149.
- Page 7, line 162: Move the references to the title of table 1.
The authors thank for this comment. The references for database have been moved next to the caption of Table 1. Please see line 210.
- Specify the type of fibers in table 1.
The authors thank for this comment. As what has been discussed in this manuscript, this study is focusing on estimating SE of carbon fibre reinforced composites. Therefore, the type of fibres involved in this study will just be carbon fibres but with varied aspect ratio.
- Page 7, line 171: there is a typo. The "fire" must be fiber.
The authors thank for this comment.
- Add more details to the paper about how you have reached the formula in equation 8.
The authors thank for this comment. The authors agree with the idea of adding more calculation steps for obtaining regression coefficients as well as biases and weightings for BPNN models. However, with the consideration of derivation steps for models might be out of the scope of this manuscript, regression model was based upon the least square method, and detailed derivation steps could be tedious and confusing, the authors finally decided to remove these derivation steps from this manuscript.
- Equation 8; modify the sign "±". It must be either "+" or "-"
The authors thank for this comment. According to the regression results, the coefficient here is supposed to be negative and the equation 8 has been corrected. Please see Equation 9 line 307.
Reviewer 3 Report
Comments on sensors-2190823
The manuscript entitled “Prediction models of shielding effectiveness of carbon fibre reinforced cement-based composites against electromagnetic interference” aims to establish an experimental database by collecting data from previous literature for shielding effectiveness of carbon fibre reinforced composites within a given frequency band.
However, the manuscript has several issues, and amendments are required to be resolved. The novelty of their work is not very clear. There are several other issues which are disclosed below:
· The abstract is really lengthy, please trim it and make it precise. Moreover, several sentences are just repeated as the authors have already mentioned in the Introduction. Several sentences are long for example, the very first sentence of the Abstract.
· The subject is not exactly new and there are two major problems in the Introduction, which must be solved prior to possible manuscript publication. In recent years, there were many advances in proposing new techniques related to the thorough study of the EMI properties at a given frequency, thickness etc. Therefore, authors must compare their proposed technique with other techniques described in the literature by different research groups (not focusing on 1-2 groups working) adapted for the same frequency interval and make a comment on the applicability of their technique for the cases of different materials.
· The introduction has some flaws, and a more detailed novelty of their work should be clearly addressed. Thorough references should be cited in the Introduction, which is missing.
· The authors should provide more discussions on the mechanisms for performance strategies, which would be beneficial for readers to understand their significance. The discussion of the EMI shielding performance and mechanism is not clear enough.
· Several sentences are just repeated in the main results which have already been discussed in the Introduction. Moreover, the authors should emphasize their work's novelty in the introduction's last paragraph.
· The resolution of the figures is not of good quality. Please provide high-resolution figures with proper labeling.
· The authors should briefly explain why the BPNN model requires heavier computation complexity than the regression model with proper references.
· Moreover, the reviewer is not completely convinced by the proposed technique since there is a lack of details about the method of extraction and processing from raw data.
· Thorough references should be provided when the authors are studying modeling techniques. It is also recommended to recheck the format of the references since the format is not uniform.
· The manuscript contains a few misprints, lost intervals and should be corrected with this respect. Similarly, the language expression in the text needs to be carefully checked and revised. There are some grammatical mistakes.
Therefore, the submitted manuscript is not suitable to be accepted for publication in this journal.
Author Response
- The abstract is really lengthy, please trim it and make it precise. Moreover, several sentences are just repeated as the authors have already mentioned in the Introduction. Several sentences are long for example, the very first sentence of the Abstract.
The authors thank for this comment. The abstract was revised based on all the comments received from both reviewers. Please see line 9 – 22.
- The subject is not exactly new and there are two major problems in the Introduction, which must be solved prior to possible manuscript publication. In recent years, there were many advances in proposing new techniques related to the thorough study of the EMI properties at a given frequency, thickness etc. Therefore, authors must compare their proposed technique with other techniques described in the literature by different research groups (not focusing on 1-2 groups working) adapted for the same frequency interval and make a comment on the applicability of their technique for the cases of different materials.
The authors thank for this comment. The authors agree with the idea of comparing proposed model with available models or approaches from previous literature to demonstrate the bottleneck in this filed and prove the novelty of this manuscript. However, after literature survey, such existing models that is able to give estimation for SE using mix design parameters were not founded in current database and this might reflect the novelty of this paper of using such regression and BPNN models as well as mix design parameters to predict SE for carbon fibre reinforced composites. Besides, the existing models are mainly associated with relaxation time, which was not intended to be included in this study. Please also see line 86 – 111.
- The introduction has some flaws, and a more detailed novelty of their work should be clearly addressed. Thorough references should be cited in the Introduction, which is missing.
The authors thank for this comment. The key aim of this manuscript is to propose a series model that can be used to predict SE of composites containing carbon fibres by using mix design parameters which are much easier to obtain compared to traditional Debye based models. The novelty of this manuscript has been added and emphasised in section 2. Research significance. Please see line 86 – 126.
- The authors should provide more discussions on the mechanisms for performance strategies, which would be beneficial for readers to understand their significance. The discussion of the EMI shielding performance and mechanism is not clear enough.
The authors thank for this comment. In order to explain the significance of the models to be proposed by this manuscript and pros and cons of existing models. Some background information has been added in the section 2. Please see line 86 – 126. To further explain or study the correlation between each factors and shielding properties, the literature review section (section 3.2 Mix design parameters affecting shielding effectiveness) has been re-organised and revised. Please see line 155 – 199.
- Several sentences are just repeated in the main results which have already been discussed in the Introduction. Moreover, the authors should emphasize their work's novelty in the introduction's last paragraph.
The authors thank for this comment. Since some reviewers are questioning about the novelty of this manuscript, one section of Section 2. Research significance has been added. Please see line 86 – 126. The authors have also found there are several sentences repeating throughout the manuscript and they have been revised or removed.
- The resolution of the figures is not of good quality. Please provide high-resolution figures with proper labeling.
The authors thank for this comment. Figures used in previous version has caused lots of confusion and misleading. All figures then used in current version have been revised or re-generated with a minimum resolution of 300 dpi and proper labelling.
- The authors should briefly explain why the BPNN model requires heavier computation complexity than the regression model with proper references.
The authors thank for this comment. The authors agree with the idea of adding references to the statement of ‘BPNN model requires heavier computation work than regression model’. Unfortunately, there is not direct literature saying that machine learning algorithms needs more calculation than regression. However, it could be noted that during model development for this manuscript, the time required for BPNN model was indeed more than regression model due to the inclusion of cycling cross-validation.
- Moreover, the reviewer is not completely convinced by the proposed technique since there is a lack of details about the method of extraction and processing from raw data.
The authors thank for this comment. Detailed presentation of data processing and model development has now been added to section 4. Experimental database (please see line 200 -210) and section 5 (please see line 275 – 358).
- Thorough references should be provided when the authors are studying modeling techniques. It is also recommended to recheck the format of the references since the format is not uniform.
The authors thank for this comment. More references have been added in the introduction as well as the section 5. And all references involved are following Vancouver referencing style.
- The manuscript contains a few misprints, lost intervals and should be corrected with this respect. Similarly, the language expression in the text needs to be carefully checked and revised. There are some grammatical mistakes. Therefore, the submitted manuscript is not suitable to be accepted for publication in this journal.
The authors thank for this comment. The expression of this manuscript has been double check before and after the revision.
Reviewer 4 Report
Thank you for being able to evaluate the manuscript entitled "Prediction models of shielding effectiveness of carbon fiber reinforced cement-based composites against electromagnetic interference," as it is a work with an exciting proposal.
Below I leave my considerations and suggestions so the work can be considered for publication.
Abstract: In general, it summarizes the idea of the work, but I think that the authors could be a little briefer, already attacking the problem directly and presenting the findings of this work more clearly, and perhaps emphasizing how the accuracy of the proposed models brings some gain to the area.
Introduction: Although it is a little long, I think the authors address the entire context underlies the proposed research. Although I believe they could have more citations to support the mentioned points.
In the excerpt, "Although everyday EMWs do not cause harmful effects, studies have shown that at a higher intensity level, EMWs do have negative impacts on pregnant women, fetuses, and infants, etc., and prolonged exposure can also cause cancer and other heart-related issues [4]" Is this statement not contradictory? Because it is clear that, in fact, chronic exposure ends up being harmful...
Several times in the text, the authors mention, "Moreover, conducting such experiments for shielding properties is also expensive and time-consuming" ... Expensive how much? and about time? Are there studies that demonstrate this? If so, it would be interesting to cite such works so as not to leave only incomplete information in the middle of the text repeated several times.
In the sequence, the authors present the equations that describe the fundamental parameters for SE; however, the equations need to be mentioned in the text, and several appear in the text without the common formalism present in articles and books.
The experimental data are presented below:
A database containing 300 pairs of data from 10 different studies was built - Where? It would be interesting if these data were made available... or at least could be consulted.
In the sequence, the authors present the discussion of the results:
Figure 2 is presented, but the data is very confusing. There is a significant overlap of data that it is not clear which work each result belongs to, which makes it very difficult to interpret...
About figure 2(a), the authors mention that it presents or illustrates the SE distribution in different fiber contents within the frequency band, but they do not report what frequency value it represents since SE is a function of frequency. This should be reported.
The authors comment that if the points referring to higher fiber contents were outliers, the results would go against what is reported in the literature. But the reasoning is unclear; why would these be outliers if the SE values are smaller than the others? Overall this is very confusing....
Regarding figure 2(b), the same problems can be considered. It is challenging to understand the data without knowing what set of data the points represent... for example, for 0% fiber content, there are a lot of points saying that SE was between 0 and 5 dB, but at the same time, other sets show between 40 and 55 dB and then as the fiber content increases some data increase others decrease, it is difficult to interpret the way it is.
Regarding the water content in Figure 3, the data presented in figure 3(a) are less confusing and show the trend described by the authors.
Since graph 3(b) is a 3D graph and with the amount of data and the chosen angle of the figure, it is a little confusing, I suggest that the authors maybe try to place the data sets separated by different colors to each work or even another parameter that differentiates the data sets. For the interpretation of trends, it would perhaps be more didactic if the authors added arrows showing the mentioned trends.
As for the sand content in the composition, it would be interesting if the authors confirmed whether the ratio between sand and cement is given in mass or volume and if these data were standardized before entering the database.
In addition, these figures present the same problem as the others, which are difficult to interpret.
In the "Thickness of specimen" part, the authors do not discuss what the observed results mean and although they mention that there is a wide range, what is observed is a large void between 15 and 100 mm thick (Figure 5(a))
The caption of Figure 5 presents an error; the data are not a function of the sand/cement ratio but of the thickness of the sample.
In figure 6, the same problem is observed, and it is unclear how the authors reached such a conclusion from these data.
Figure 6 also has an error in the caption...
In item 4, "models of prediction," it would be interesting if the authors made available the models that were developed in MATLAB.
In the sequence, the authors present the regression model and the values of each of the obtained coefficients, but it is not clear if other models were tested; the authors could also have plotted this model to show how well this model fits the data of the proposed bank with the different parameters.
Finally, the authors assess the quality of the model using R2 and RSME but do not mention whether the other assumptions for obtaining the model were verified.
Conclusion:
In general, the authors point out the main findings of the manuscript, but it would be interesting if each of these conclusions had been better discussed in the results and discussion.
In addition, the authors could conclude how their models allowed the selection of ideal formulations for obtaining the minimum SE requirements in the 1kHz to 1.5GHz range, as the US Department of Defense desired.
Or what are the necessary or minimum requirements that the models allow stipulating to improve the shielding properties and correlate them with the electrical conductivity and the permeability of the material, factors that are so important in the introduction but that are no longer discussed throughout the work
Author Response
- Abstract: In general, it summarizes the idea of the work, but I think that the authors could be a little briefer, already attacking the problem directly and presenting the findings of this work more clearly, and perhaps emphasizing how the accuracy of the proposed models brings some gain to the area.
The authors thank for this comment. The abstract was revised based on all the comments received from both reviewers. Please see line 9 – 22.
- Introduction: Although it is a little long, I think the authors address the entire context underlies the proposed research. Although I believe they could have more citations to support the mentioned points.
The authors thank for this comment. The introduction has been re-organised and revised based on all the comments received from both reviewers regarding this section. A few more references have also been added to this section and other sections.
- In the excerpt, "Although everyday EMWs do not cause harmful effects, studies have shown that at a higher intensity level, EMWs do have negative impacts on pregnant women, fetuses, and infants, etc., and prolonged exposure can also cause cancer and other heart-related issues [4]" Is this statement not contradictory? Because it is clear that, in fact, chronic exposure ends up being harmful...
The authors thank for this comment. The sentence mentioned is indeed contradictory and has been removed from the context.
- Several times in the text, the authors mention, "Moreover, conducting such experiments for shielding properties is also expensive and time-consuming" ... Expensive how much? and about time? Are there studies that demonstrate this? If so, it would be interesting to cite such works so as not to leave only incomplete information in the middle of the text repeated several times.
The authors thank for this comment. The authors believe that the original meaning of this sentence is saying performing numerical modelling is indeed much cheaper than conducting physical experiments. The authors have also been aware of the improper use of this kind of misleading statement and all these statements have been revised accordingly.
- In the sequence, the authors present the equations that describe the fundamental parameters for SE; however, the equations need to be mentioned in the text, and several appear in the text without the common formalism present in articles and books.
The authors thank for this comment. Some explanation or introduction for such parameters have been added to the context.
The experimental data are presented below:
- A database containing 300 pairs of data from 10 different studies was built - Where? It would be interesting if these data were made available... or at least could be consulted.
The authors thank for this comment. The database contains 346 pairs of data in total and covers data including fibre content (%, in volume), water-to-cement ratio (by mass), sand-to-cement ratio (by mass), aspect ratio of carbon fibre, specimen thickness (mm), frequency (GHz), and shielding effectiveness (dB). The whole database has been attached to the end of the manuscript in the appendix and been highlighted in yellow colour. Please see line 200 – 210 and 511 – 513.
In the sequence, the authors present the discussion of the results:
- Figure 2 is presented, but the data is very confusing. There is a significant overlap of data that it is not clear which work each result belongs to, which makes it very difficult to interpret...
The authors thank for this comment. Since measured overall SE is also dependent on frequency, therefore, the authors initially aimed to present the scatter in a 3D plot combining overall SE and fibre content. However, as what was reported by all the reviewers, this kind of 3D plot could not be able to present the data in any sufficient form. The authors therefore removed all 3D plots and replaced with 2D scatter plots with the third variable being involved as colour scale. Please see line 229 – 230.
- About figure 2(a), the authors mention that it presents or illustrates the SE distribution in different fiber contents within the frequency band, but they do not report what frequency value it represents since SE is a function of frequency. This should be reported.
The authors thank for this comment. The authors agree with the statement of overall SE is dependent on frequency. Combining the problem with 3D scatters, the original figures were replaced with 2D scatters with frequency being set as colour scale. Please see line 229 – 230.
- The authors comment that if the points referring to higher fiber contents were outliers, the results would go against what is reported in the literature. But the reasoning is unclear; why would these be outliers if the SE values are smaller than the others? Overall this is very confusing....
The authors thank for this comment. The authors believe that the confused observations or statement are mainly attributed to the interpretation of previous 3D figures. In this revised manuscript, all the discussion involved and figures associated with are revised. Please see line 200 – 274.
- Regarding figure 2(b), the same problems can be considered. It is challenging to understand the data without knowing what set of data the points represent... for example, for 0% fiber content, there are a lot of points saying that SE was between 0 and 5 dB, but at the same time, other sets show between 40 and 55 dB and then as the fiber content increases some data increase others decrease, it is difficult to interpret the way it is.
The authors thank for this comment. The authors believe that the confused observations or statement are mainly attributed to the interpretation of previous 3D figures. In this revised manuscript, all the discussion involved and figures associated with are revised. Please see line 200 – 274.
- Regarding the water content in Figure 3, the data presented in figure 3(a) are less confusing and show the trend described by the authors.
The authors thank for this comment. The authors believe that the confused observations or statement are mainly attributed to the interpretation of previous 3D figures. In this revised manuscript, all the discussion involved and figures associated with are revised. Please see line 200 – 274.
- Since graph 3(b) is a 3D graph and with the amount of data and the chosen angle of the figure, it is a little confusing, I suggest that the authors maybe try to place the data sets separated by different colors to each work or even another parameter that differentiates the data sets. For the interpretation of trends, it would perhaps be more didactic if the authors added arrows showing the mentioned trends.
The authors thank for this comment. The authors have been aware of the misleading figures used in previous manuscript and appreciate for the idea of introducing the third variable in colour scale. The figures associated with this type of problem have been removed and replaced by 2D scatters with colour scale. Please see line 200 – 274.
- As for the sand content in the composition, it would be interesting if the authors confirmed whether the ratio between sand and cement is given in mass or volume and if these data were standardized before entering the database.
The authors thank for this comment. The ratio of sand to cement is in mass ratio and fibre content is volume percentage. Data gathered and processed for development are raw data without normalisation. Please see line 207 – 209.
- In addition, these figures present the same problem as the others, which are difficult to interpret.
The authors thank for this comment. The authors believe that the confused observations or statement are mainly attributed to the interpretation of previous 3D figures. In this revised manuscript, all the discussion involved and figures associated with are revised. Please see line 200 – 274.
- In the "Thickness of specimen" part, the authors do not discuss what the observed results mean and although they mention that there is a wide range, what is observed is a large void between 15 and 100 mm thick (Figure 5(a))
The authors thank for this comment. The authors agree with the question of the existence of big blank in between 10 mm and 100 mm. And the authors believe that it is mainly coming from the papers involved used different standards but most of them are falling around 10 mm. As the previous observation is not accurate and convincible, the authors have revised the results discussion regarding this section. Please see line 256 – 267.
- The caption of Figure 5 presents an error; the data are not a function of the sand/cement ratio but of the thickness of the sample.
The authors thank for this comment. The figure involved has been removed and replaced by new figure.
- In figure 6, the same problem is observed, and it is unclear how the authors reached such a conclusion from these data.
The authors thank for this comment. The authors agree with the question as stated above and this previous conclusions regarding this section have been revised.
- Figure 6 also has an error in the caption...
The authors thank for this comment. The figure involved has been removed and replaced by new figure.
- In item 4, "models of prediction," it would be interesting if the authors made available the models that were developed in MATLAB.
The authors thank for this comment. The models proposed by this paper were available in the paper under section 4.1 Regression model (please see line 307) and 4.2 BPNN model (please see line 355 – 358).
- In the sequence, the authors present the regression model and the values of each of the obtained coefficients, but it is not clear if other models were tested; the authors could also have plotted this model to show how well this model fits the data of the proposed bank with the different parameters.
The authors thank for this comment. The authors agree with the idea of comparing proposed model with available models or approaches from previous literature to demonstrate the bottleneck in this filed and prove the novelty of this manuscript. However, after literature survey, such existing models that is able to give estimation for SE using mix design parameters were not founded in current database and this might reflect the novelty of this paper of using such regression and BPNN models as well as mix design parameters to predict SE for carbon fibre reinforced composites. Besides, the existing models are mainly associated with relaxation time, which was not intended to be included in this study. Please see line 86 – 111.
- Finally, the authors assess the quality of the model using R2 and RSME but do not mention whether the other assumptions for obtaining the model were verified.
The authors thank for this comment.
Conclusion:
- In general, the authors point out the main findings of the manuscript, but it would be interesting if each of these conclusions had been better discussed in the results and discussion.
The authors thank for this comment. According to all the comments received, the conclusion section has been revised as line 360 – 409.
- In addition, the authors could conclude how their models allowed the selection of ideal formulations for obtaining the minimum SE requirements in the 1kHz to 1.5GHz range, as the US Department of Defense desired.
The authors thank for this comment. The authors agree with the idea of further interpreting the possible use of the proposed models or the novelty of this manuscript from the view of certain possible stakeholders. It will be fairly interesting to have such discussion in a paper, however, it might be out of scope of a single research paper.
- Or what are the necessary or minimum requirements that the models allow stipulating to improve the shielding properties and correlate them with the electrical conductivity and the permeability of the material, factors that are so important in the introduction but that are no longer discussed throughout the work.
The authors thank for this comment. The authors agree with the idea of emphasising the impact of such factors, e.g., permeability or conductivity. However, as what has been mentioned in this manuscript, the aim of this paper is developing a kind of mathematical model which does not involve any of these indirect factors but using parameters that can be easily obtained without performing any physical tests to estimate the shielding effectiveness against EMW. Please also see line 86 – 111.
Round 2
Reviewer 3 Report
The authors have answered all the queries. The reviewer has no further questions to ask. The manuscript can be accepted for publication. Please check the format. There are some issues observed in the reviewers' pdf file. Needs to be acknowledged.
Author Response
Thanks for the comments.
Reviewer 4 Report
I want to congratulate the authors for their effort in providing the changes proposed by the reviewers to improve the quality of the submitted manuscript.
The modifications added more quality to work, but some points need a little more effort to make the manuscript clearer and more attractive.
Most of the questions raised were answered or modified satisfactorily. However, some points may not have been so clear.
About 5 answers, "The authors thank you for this comment. Some explanation or introduction for such parameters has been added to the context." Perhaps it was not clear what I suggested, but although the modification has improved, I believe it is still necessary to correlate the Equation in the text. For example Line 93, “Its frequency-dependent complex relative permittivity obeys the Equation 1 [10]:”
Regarding 6 answers, I congratulate the authors on the changes. I suggest that in the table added in the appendix, as these are data from the literature, it is essential that each of the data presented is referenced with the respective work for consultation.
I also suggest that the authors try to revise the text again, as some points about English have become more confusing, in addition to formatting errors in this new version.
On line 233, a poorly formatted text appears; in the sequence, figure 3 appears on line 239 and is repeated on line 243. The exact repetition of figures occurs in the others.
Regarding the dispersion results presented, I agree with the modification made by the authors, aiming to make it less confusing, but even so, it is difficult to understand the trends in the data presented. For example, it is difficult to understand Figure 2. The data presented between 0 and 1% of fibre present values from 0 to 70 dB without showing any tendency, only a few points show a higher fibre content, but in general, the observed values are restricted to much smaller values than those observed for materials between 0 and 1%.
The explanation of this observed trend would be valid only if we disregard the whole range of data showing much higher values for smaller fibre concentrations.
A figure by itself needs to be self-explanatory, and the data needs to make clear the trends to watch. As data vary greatly, I suggest authors review the input data used in this work. Because it makes no sense to present figures that cannot be clearly understood, or worse, the results do not express any real trend. Because in that case, the figure becomes just a pile of meaningless data.
Author Response
- About 5 answers, "The authors thank you for this comment. Some explanation or introduction for such parameters has been added to the context." Perhaps it was not clear what I suggested, but although the modification has improved, I believe it is still necessary to correlate the Equation in the text. For example Line 93, “Its frequency-dependent complex relative permittivity obeys the Equation 1 [10]:”
The authors thank for this comment. Please see line 93, 132 – 134, 140 – 141, 142 – 144, 147 – 148, 149 – 151, 152 – 154.
- Regarding 6 answers, I congratulate the authors on the changes. I suggest that in the table added in the appendix, as these are data from the literature, it is essential that each of the data presented is referenced with the respective work for consultation.
The authors thank for this comment. The references for each individual data set were added and please see line 518 – 520.
- I also suggest that the authors try to revise the text again, as some points about English have become more confusing, in addition to formatting errors in this new version.
The authors thank for this comment.
- On line 233, a poorly formatted text appears; in the sequence, figure 3 appears on line 239 and is repeated on line 243. The exact repetition of figures occurs in the others.
The authors thank for this comment. The poorly formatted captions have been revised and repetitions of figures have been revised or removed. Please see line 231 – 234, 242 – 244, 250 – 252, 262 – 263, 273.
- Regarding the dispersion results presented, I agree with the modification made by the authors, aiming to make it less confusing, but even so, it is difficult to understand the trends in the data presented. For example, it is difficult to understand Figure 2. The data presented between 0 and 1% of fibre present values from 0 to 70 dB without showing any tendency, only a few points show a higher fibre content, but in general, the observed values are restricted to much smaller values than those observed for materials between 0 and 1%. The explanation of this observed trend would be valid only if we disregard the whole range of data showing much higher values for smaller fibre concentrations.
The authors thank for this comment. In order to further minimise the confusion or misleading caused by the figures in Section 4 Experimental database, especially Figure 2 Scatter plot of SE against fibre content, the authors then have remade the Figure 2 into three subfigures by splitting the figure into three by fibre content of 0 – 4%, 0 – 1%, and 1 – 4% (beyond 1 %), please also see line 235 – 236. Figure 2 (C) reveals a obvious tendency of increasing fibre content improves SE at both low or high frequency bands while from Figure 2 (b) the trend seems not clear or obvious. This might be attributed to
- Only few data points are beyond 1% and they are from the same literature, meaning most of data are lying below the threshold of 1%. This also indicates the research gap of shielding properties with high fibre content as well as at high frequency band. If more data are available in the future, the data aggregation might be able to show a much clearer trend or tendency.
- The shielding properties are governed by multiple mechanisms and a few other factors are also affecting them. Since this study is based upon the data gathered from previous literature, the effect of single parameter cannot be fully explained without eliminating the effects of other parameters or mechanisms. The authors believe that using statistical method such as variance, co-variance, etc are not meaningful to interpretate the data since the database itself might not satisfy certain statistical assumption, such as lack of rank.
- A figure by itself needs to be self-explanatory, and the data needs to make clear the trends to watch. As data vary greatly, I suggest authors review the input data used in this work. Because it makes no sense to present figures that cannot be clearly understood, or worse, the results do not express any real trend. Because in that case, the figure becomes just a pile of meaningless data.
The authors thank for this comment. The authors agree with the point of a figure is expected to be self-explanatory or at least showing clear tendency or obvious pattern and have noticed that some figures, especially Figure 2 seemed complicated to see and comprehend. On the other hand, this unclear trend reflects that the shielding mechanisms are complicated and effects of parameters including w/b, s/c, fibre type, or specimen thickness, etc, are not negligible and indicating the potential of performing experiments by control variates method to further investigate the effect of fibre content on SE solely.